# Exploring the Role of IL-36 Cytokines as a New Target in Psoriatic Disease

**DOI:** 10.3390/ijms22094344

**Published:** 2021-04-21

**Authors:** Helena Iznardo, Lluís Puig

**Affiliations:** 1Dermatology Department, Hospital de la Santa Creu i Sant Pau, 08041 Barcelona, Spain; hiznardo@santpau.cat; 2Department of Medicine, School of Medicine, Universitat Autònoma de Barcelona, 08041 Barcelona, Spain

**Keywords:** psoriasis, pustular psoriasis, psoriatic arthritis, imsidolimab, spesolimab, IL-36, IL-36R, pathogenesis

## Abstract

Unmet needs in the treatment of psoriasis call for novel therapeutic strategies. Pustular psoriasis and psoriatic arthritis often represent a therapeutic challenge. Focus on IL-36 cytokines offers an interesting approach, as the IL-36 axis has been appointed a critical driver of the autoinflammatory responses involved in pustular psoriasis. Two IL-36R blocking antibodies, imsidolimab and spesolimab, are currently undergoing phase II and III clinical trials, with promising results.

## 1. Introduction

Psoriasis is a chronic and recurrent immune-mediated disease that affects approximately 3% of the US population, with a similar prevalence in European countries [1]. The most common presentation is plaque psoriasis, which is characterized by scaly, sharply demarcated plaques affecting extensor surfaces, often with a symmetrical distribution. Pustular psoriasis is rare and presents with sterile, neutrophil-rich pustules that can be localized or generalized [2]. Pustular psoriasis phenotypes include generalized pustular psoriasis (GPP), palmoplantar pustular psoriasis (PPP), and acrodermatitis continua of Hallopeau (ACH) [3]. Association between pustular psoriasis and plaque psoriasis is common, with patients often presenting with pustules and plaques altogether. However, clinical, histopathological, and genetic differences suggest that these two might be different entities [3,4,5,6]. 

Psoriasis is a multisystem disorder with a great impact on quality of life and important comorbidity. Psoriatic arthritis (PsA) is a complex inflammatory joint disease included in the spondyloarthropathy spectrum that affects approximately one-third of psoriasis patients, especially those with moderate-to-severe psoriasis [7]. Psoriasis symptoms precede PsA in 85% of patients, although they can occur simultaneously or, more rarely, be followed by cutaneous symptoms [8]. PsA produces stiffness, pain, and swelling of joints, and it can progress to debilitating joint destruction. Enthesitis and dactilitys are observed in 30 to 50% and 40 to 50% of patients, respectively [9]. Clinical presentation is variable, and different patterns of involvement have been described: oligoarticular, polyarticular, distal, arthritis mutilans, and axial [10]. These may change over time within the same patient [11]. 

The pathogenesis of psoriasis is not completely understood. Genetic, epigenetic, and non-genetic factors—mainly the microbiome and environmental factors—altogether participate in psoriasis and PsA susceptibility [12]. The innate and adaptive immune systems play a critical role, especially CD4 and CD8 cells and the interleukin (IL)-23/IL-17 immune axis [13]. More recently, the identification of specific mutations in the IL36RN gene encoding the IL-36 receptor antagonist (IL-36Ra) have been linked to an autoinflammatory condition characterized by a severe form of GPP called “deficiency of interleukin-36-receptor antagonist” (DITRA); this highlights the importance of IL-36 family cytokines as critical mediators of psoriatic disease [14,15]. Other genes with mutations associated with pustular psoriasis include caspase-activating recruitment domain member 14 (CARD14), adaptor protein complex 1 subunit sigma 3 (AP1S3), TNFAIP3-interacting protein 1 (TNIP1), and serpin family A member 3 (SERPINA 3) [16]. All of them are involved in IL-1/IL-36 signaling pathways, further underscoring the relevance of these cytokines in pustular psoriasis. 

Phenotype/genotype correlation in pustular psoriasis is complicated, and controversial results have been published. IL36RN mutations are the genetic variant most frequently observed in patients with pustular psoriasis (5–24%), followed by AP1S3 in 7–11% and CARD14 in a very small number of subjects [17]; the highest rates correspond to patients with GPP, and the lowest correspond to PPP in all cases. While earlier reports found earlier ages of onset and higher risks of systemic inflammation in patients with IL36RN mutations [14], more recent studies have shown no difference in disease severity between patients with IL36RN single heterozygous mutations and homozygous or compound heterozygous mutations [18,19]. To further complicate this matter, a significant association of IL-36RN mutations with early age of onset, regardless of phenotype, was found in a recent study including 863 patients with pustular psoriasis [17,20]. The percentage of patients harboring IL36RN mutations varies among studies, and different reports indicate that the sites of mutation differ among ethnicities. In addition, the mutation rates differ among pustular psoriasis phenotypes. In a study on 57 Chinese patients with pustular psoriasis, IL36RN mutations were found in 75% of GPP patients and 94% of ACH patients [21]. On the contrary, Takahashi and colleagues performed mutation analysis of the IL36RN gene in 88 Japanese patients with PPP and identified three types of single base substitutions of IL36RN; they were heterozygous and different from those found in European studies, and they were considered to be of no pathogenic relevance [22,23]. Furthermore, other studies in patients with PPP have shown that the combined frequency of AP1S3 and IL36RN mutations accounts for less than 10% of patients, suggesting not only that PPP is not clearly associated with IL36RN but also that known genes account for only a minority of disease cases [17,22]. 

Many advancements have been made in psoriasis treatment in this century. Identification of pro-inflammatory cytokines such as tumor necrosis factor (TNF)α, IL-17, and IL-23 as therapeutic targets allowed the development of several anti-cytokine monoclonal antibodies that revolutionized treatment of psoriasis [8]. Nevertheless, unmet needs remain, since some patients do not respond, stop responding over time, or suffer from side effects of these treatments. This especially applies to pustular psoriasis and PsA, where resistance to existing treatments and disease recurrence are commonplace. Therefore, novel therapeutic strategies are needed. The blockade of IL-36 family cytokines represents an interesting therapeutic target for psoriasis, pustular psoriasis, and PsA, among other diseases, and this will be the subject of this PubMed search strategy “IL-36” AND (“psoriasis” OR “pustul*”)-based review. 

## 2. The Role of IL-36 Family Cytokines in Psoriasis Pathogenesis

IL-36 cytokines belong to the IL-1 family and play an important role in the maintenance of endogenous homeostasis. They act as regulators of the innate immune system, and their uncontrolled activation and expression results in pathologic inflammatory responses, including psoriasis [24]. IL-36 cytokines are expressed in epithelial and immune cells and comprise three agonists with pro-inflammatory functions (IL-36α, IL-36β, and IL-36γ) and two antagonists (IL-36RN or IL-36Ra and IL-38); all of them use the same receptor complex IL-36R [25] (Figure 1). IL-36α and IL-36β are normally present in healthy skin, whereas IL-36γ is present at high levels in psoriasis lesions [25]. In the skin, IL-36α and IL-36γ are predominantly produced by epidermal keratinocytes (KCs) but also by dermal fibroblasts, endothelial cells, macrophages, Langerhans cells (LCs), and dendritic cells (DCs) [25]. KCs are the main producers of IL-1Ra and IL-38 in resident skin cells [24]. Overexpression of IL-36α, IL-36β, and IL-36γ has been found in the skin and serum of psoriasis patients with a positive correlation between disease severity and cytokine levels [26,27]. Furthermore, IL-38 levels are reduced in psoriasis patients, highlighting the unbalance of IL-36 axis in favor of pro-inflammatory IL-36 agonists [28]. 

IL-36 cytokines are released as precursors and require N-terminal truncation to acquire their full pro-inflammatory activity, but—unlike other IL-1 cytokines—they do not contain a caspase cleavage site [29]. Processing and activation are produced by proteases: cathepsin G, proteinase 3, and elastase (neutrophil-derived), and cathepsin S (derived from KCs and fibroblasts) [30,31,32,33,34]. Neutrophil-derived proteases are found in neutrophil extracellular traps (NETs) resulting from the programmed death of neutrophils [32]. Protease inhibitors secreted by KCs can regulate the IL-36-mediated inflammatory cascade. Alpha-1-antitrypsin and alpha-1-antichymotrypsin, codified by SERPINA1 and SERPINA3 genes, inhibit the processing of IL-36 cytokines by neutrophil elastase and cathepsin G, respectively [5]. 

Upon binding to their receptor, IL-36 cytokines induce the activation of nuclear factor-κβ (NF-κβ) and mitogen-activated protein kinases, leading to the promotion of pro-inflammatory responses in KCs, fibroblasts, dermal endothelial cells, macrophages, DCs, and various T cell subsets [35]. Activated IL-36 cytokines induce the expression of pro-inflammatory cytokines, chemokines, and co-stimulatory molecules cells—such as IL-1β, IL-12, IL-23, IL-6, and TNF-α, CCL1, CXCL1, and GM-CSF—by DCs and Th1 lymphocytes [36]. They also induce T-cell proliferation, with further potentiation of the inflammatory response, and signal to KCs in an autocrine manner, stimulating the additional production of pro-inflammatory cytokines, antimicrobial peptides, and neutrophil chemokines (CXCL1, CXCL2, and CXCL8) [37,38]. Cytokines secreted by infiltrating Th1 and Th17 lymphocytes further potentiate this inflammatory loop by inducing expression of IL-36 and other pro-inflammatory mediators such as TNF-α, IL-6, and CXCL8 by KCs [39] (Figure 2).

IL-36-driven inflammation is mainly orchestrated by KCs, as demonstrated recently by Hernández-Santana and colleagues [40]. These authors found that targeted deletion of IL-36R in KCs resulted in similar levels of protection from psoriasiform inflammation as those observed in IL-36R-deficient mice. IL-36R-deficient mice KCs also featured significantly decreased expression of IL-17A, IL-23, and IL-22, as well as loss of chemokine-induced infiltration of the inflamed skin by neutrophils and IL-17A—expressing γδ T-cell subsets [40]. These data suggest that IL-36-derived signaling in KCs play an early role in the initiation and self-amplification of the “feed-forward” model of psoriatic inflammation through directly regulating IL-17A expression [40]. 

IL-36 cytokines also enhance psoriatic skin inflammation by activating angiogenesis and promoting leukocyte recruitment. IL-36α and IL-36γ induce KCs expression and the release of proliferative factors—such as heparin-binding EGF-like growth factor (HB-EGF) and vascular endothelial growth factor (VEGF)-A—acting on fibroblasts and endothelial cells [28]. In addition, dermal endothelial cells activated by IL-36γ show increased release of IL-6, CXCL8, CCL20, and CXCL1, increased expression of adhesion molecules, and increased proliferation and branching [28,41]. 

Other immune cells expressing the IL-36 receptor, such as macrophages, LCs, and DCs, are also activated by IL-36 cytokines. IL-36γ-stimulated macrophages produce high levels of IL-23 and TNFα, activating endothelial cells and leading to increased adherence of monocytes. In turn, monocytes increase the release of IL-23, promoting the polarization of IL-17/IL-22-expressing lymphocytes [41]. In addition, anti-inflammatory M2 macrophages are driven to a pro-inflammatory M1 phenotype [36]. High levels of IL-36R mRNA have been found in LCs and dermal CD1a DCs. IL-36 cytokines potentiate Toll-like-receptor (TLR)-9 activation and IFNα production, contributing to extracutaneous inflammation in psoriasis [42] They are also quintessential in eliciting generalized, acute response inflammation. Last, although neutrophils do not express IL-36R, IL-36 cytokines indirectly contribute to their activation by inducing the release of neutrophil chemoattractants by KCs, as previously noted. NETs are released by neutrophils in a process called NETosis, with three possible models: (a) suicidal NETosis—dependent on production of reactive oxygen species (ROS) via the Mitogen-Activated Protein Kinases (MEK)-extracellular signal-regulated kinases (ERK) signaling pathway, (b) TLR and complement receptor for C3 protein-related NETosis, and (c) mitochondrial ROS production-dependent NETosis. High numbers of NETotic cells have been found in the peripheral blood of psoriasis patients, and they were correlated with disease severity [43].

The innate immune response has a predominant role in the immunopathogenesis of pustular psoriasis, with activation of the IL-1/IL-36 inflammatory axis leading to neutrophil chemotaxis and neutrophil-driven inflammatory responses [16]. Mutations in genes coding for IL-36 axis components—mainly the IL-36RN gene but also the SERPINA3 gene—and NFκB signaling pathways (CARD14, AP1S3, TNIP1) have been associated with both GPP and palmoplantar/acral pustular psoriasis [14,44]. Moreover, findings of microarray analysis studying transcriptomes from patients with plaque psoriasis and GPP reflect this IL-1/IL-36 protagonism: expression of IL-1β, IL-36α, IL-36γ, and neutrophil chemokines (CXCL1, CXCL2, CXCL8) was higher, while IL-17A and IFNγ expression was lower in GPP lesions as compared to plaque psoriasis lesions [5]. Arakawa and colleagues studied the role of IL-36 cytokines in linking the innate immune system activation and T-cell responses in GPP. They found a broad activation of CD4+ T-cells with increased IL-17A and TNF production in blood and skin lesions of GPP patients, as well as increased infiltration of IL-17A+ cells in skin lesions from GPP as compared to plaque psoriasis patients [45]. Inmunohistochemical studies revealed that lesional CD4+ T cells expressed IL-36R and hyperproliferated after IL-36β stimulation [45]. Furthermore, antigen-driven CD4+ T-cell responses were present in GPP patients irrespective of their IL-36RN mutation status [45]. 

IL-36α is expressed and produced by macrophages and plasma cells in the synovium of PsA patients, with expression levels similar to those in rheumatoid arthritis (RA) [46,47]. However, Boutet and colleagues found a deficient expression of IL-36Ra and IL-38 in comparison to RA, permitting an easier activation of downstream pro-inflammatory cascades by IL-36α in PsA [47]. Evidence of IL-36 activation in PsA was underpinned by a higher expression of cathepsin G and neutrophil elastase in PsA [47]. In addition, the authors also found higher responsiveness of fibroblast-like synoviocytes (FLS) derived from inflamed psoriatic joints to IL-36α stimulation, with higher production of IL-8 and stronger NF-κβ activation in PsA-FLS as compared to RA-FLS. TNFα stimulation caused a comparable production of pro-inflammatory molecules in both disease models, suggesting a disease-dependent response specific for IL-36 [47]. 

IL-38 is a 17–18 kDa protein that shares 40% sequence similarity with IL-1Ra and IL-36Ra (antagonists of IL-1 and IL-36, respectively) and binds IL-36R to antagonize IL-36 [15]. IL-38 is expressed mostly in the skin and immune cells, and its expression is downregulated by inflammatory cytokines [28,48]. IL-17, IL-22, and IL-36γ have inhibitory effects on IL-38 expression, although they induce the expression of IL-36Ra. IL-38 has a role in KC differentiation: its expression is reduced in de-differentiated KCs, whereas terminally differentiated KCs release higher levels of IL-38 relative to IL-36Ra [8]. IL-38 can suppress the production of IL-17A by γδ T-cells through IL-1RAcP antagonism [48].

## 3. Targeting IL-36R in Psoriasis

The binding of IL-36Ra and IL-38 to IL-36R inhibits the activation of intracellular pathways, and the impact of IL-36 blockade in psoriasis has been widely studied. 

Several experimental models have shown potential for IL-36R inhibiting antibodies. Mice deficient in IL-36R or treated with IL-36R antibodies are protected from imiquimod (IMQ)-induced skin inflammation [49,50]. In murine models of IL-23 or IL-36α injection, IL-36R antibodies inhibit inflammatory responses, with significant attenuation of skin thickening and expression of psoriasis-relevant genes [51]. Hovhannisyan and colleagues studied a mouse model with decreased affinity of IL-36Ra to IL-36R, leading to enhanced IL-36R signaling [52]. Using IL-36R monoclonal antibodies, they demonstrated that IL-36R blockade led to the down-regulation of IL-17A, IL-17F, and IL-23, as well as to reduced skin thickness and reduced IL-6, IL-1β, and TNFα cytokines in the skin [52]. In addition, they found comparable efficacy of anti-IL36R and anti-IL-12p40 (a murine surrogate of ustekinumab) antibodies in reducing IMQ-induced skin inflammation in vivo [52]. Moreover, while anti-IL36R administration led to decreased levels of IL-36α, IL-36β, and IL-36γ; anti-IL-12p40 antibody only achieved a significant reduction of IL-36γ [52]. 

Blocking the IL-36 pathway might also be of use in PsA patients, especially in those refractory to treatment. High levels of IL-36α expression were found in synovium of patients with PsA not responding to disease-modifying anti-rheumatic drugs (DMARDs). Moreover, high expression remained unchanged after DMARDs treatment, with PsA-derived synovial fibroblasts exhibiting higher production of IL-36 agonists with down-regulation of antagonists and other pro-inflammatory cytokines such as IL-1β and TNFα. Finally, treatment with IL-36Ra was associated with down-regulation of CXCL8 production by PsA-FLS [47].

Regarding safety, genetic studies with individuals carrying loss-of-function mutations in the IL-36R gene show that immune function is preserved in these patients, suggesting that IL-36 can be targeted without compromising host defense [53]. 

Altogether, these findings support the clinical development of IL-36R blocking antibodies, and currently, there are two molecules undergoing phase II and III clinical trials. 

Spesolimab (BI655130, Boehringer Ingelheim) has demonstrated efficacy in a phase I, open-label, proof-of-concept study of seven biologic-naïve adult patients with a moderate GPP flare. All patients treated exhibited rapid skin improvement after the administration of a single intravenous dose of 10 mg/kg of spesolimab [54]. A 73.2% and 79.8% Generalized Pustular Psoriasis Area and Severity Index reduction was observed at 2 and 4 weeks, respectively, and it was maintained at week 20 [54]. No serious adverse events were reported. Spesolimab was effective regardless of the presence of GPP-associated mutations (three patients had homozygous IL-36RN mutations and one had a heterozygous CARD14 mutation) [54]. Molecular response was characterized by the rapid downregulation of genes involved in the innate immune response, Th1/Th17-mediated inflammation and neutrophilic pathways, both in the skin and serum of treated patients [54]. Phase II and III studies of spesolimab are currently being performed in GPP, palmoplantar pustulosis, and other potential indications under study, including ulcerative colitis, Crohn’s disease, and atopic dermatitis [55]. 

An ongoing Phase II study of imsidolimab (ANB019, AnaptysBio) with GPP and PPP patients will assess the efficacy and safety of this anti IL-36R monoclonal antibody. Reported results of a phase I study suggest a favorable side effect profile [56]. 

Other therapeutic approaches targeting IL-36 signaling pathways are under study. IL-36y blockade with a small IL-36y-binding molecule has shown inhibition of inflammation, both in vitro and in vivo, through the blockade of interactions with IL-36R [57]. Antibodies directed against IL-1RAcP, which is highly expressed in chronic myeloid leukemia stem cells, have been developed and evaluated in pre-clinical disease models of chronic myeloid leukemia and acute myeloid leukemia [58]. Since IL-1RAcP is commonly expressed by various cell and tissue types, more studies are needed in order to qualify this molecule as a safe target for treatment of psoriasis.

## 4. Conclusions

Better understanding of the detailed molecular mechanisms of psoriasis has allowed the development of novel molecular-targeted therapies for treating psoriasis more effectively. Focus on the IL-23/IL-17 axis and Th17-related cytokines TNF, IL-23, and IL-17 has provided monoclonal antibodies that mitigate skin lesions and related symptoms, including irreversible joint destruction caused by PsA. 

Unmet needs are especially relevant in the management of pustular psoriasis and PsA patients, and they call for further research on psoriasis pathogenesis. The IL-36 axis is a critical driver of the autoinflammatory responses involved in pustular psoriasis, with IL-36 cytokine hyperactivation resulting in the promotion of neutrophil chemotaxis and neutrophil-driven inflammatory responses. Moreover, the participation of IL-36 cytokines in PsA has also been explored, and more research is needed to highlight their relevance on the synovial inflammation. Monoclonal antibodies targeting IL-36R have shown promising results in patients with GPP and could also be useful in PsA. Ongoing Phase II and III clinical trials with the IL-36R blocking antibodies imsidolimab and spesolimab in GPP and PPP patients will probably confirm these results, but the initial reports on PPP have been somewhat disappointing [59]. Whether IL-36-inhibitors are effective following the failure of IL-17- or IL-23-inhibitors remains to be demonstrated with the inclusion of bio-experienced patients in ongoing clinical trials. 

Studies on the different IL-36 isoforms, as well as their regulators (IL-36Ra, IL-38, and protease inhibitors), and their specific roles on the IL-36 pro-inflammatory axis and the pathogenesis of psoriatic disease are a research priority. Understanding the biological contribution of these molecules to psoriasis, pustular psoriasis and PsA will allow the design of targeted therapies and personalized medicine strategies for these diseases. 

## Figures and Tables

**Figure 1 ijms-22-04344-f001:**
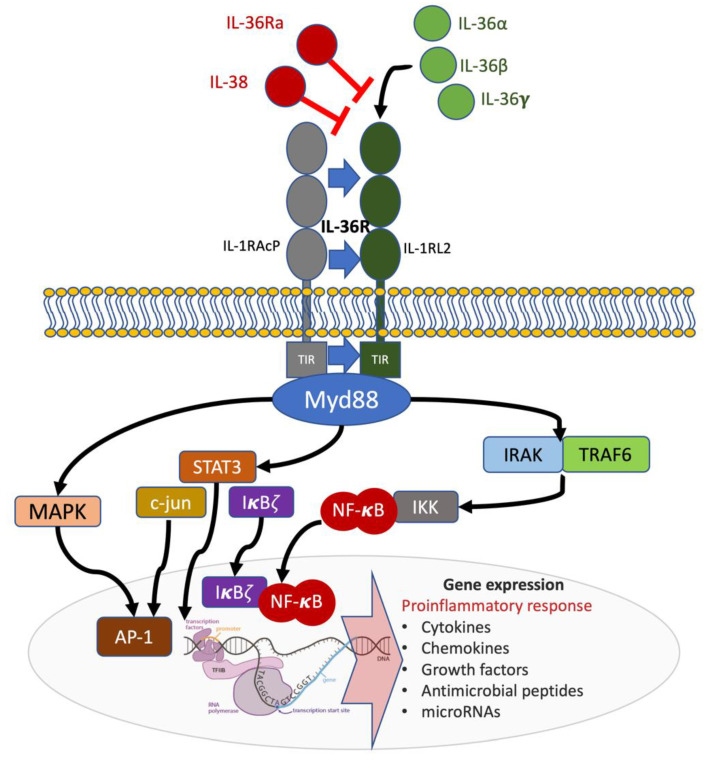
Receptor and signaling pathways involved in the IL-36 axis. IL-36R is a heterodimeric complex composed of an IL-1RL2 subunit and IL-1RAcP co-receptor. Agonist binding (IL-36α, IL-36β or IL-36γ) to IL-1RL2 results in IL-1RAcP recruitment and the activation of intracellular pathways. The MyD88/IRAK1/IRAK2/TRAF6 platform with corresponding intracellular signaling results in pro-inflammatory gene expression. Antagonism by IL-36Ra and IL-38 inhibits IL-36R signaling (shown in red). TIR: Toll/interleukin-1 receptor; MyD88: Myeloid differentiation primary response (88); NF-kB: nuclear factor kB; IRAK: Interleukin 1 receptor-associated kinase; TRAF: TNF receptor-associated factor; STAT3: Signal transducer and activator of transcription 3.

**Figure 2 ijms-22-04344-f002:**
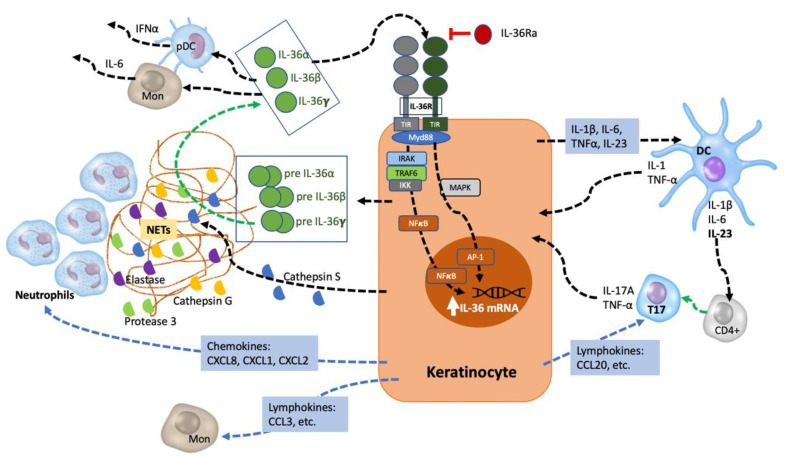
IL-36 autocrine and autoinflammatory circuits. IL-1, TNF, and IL-17A induce IL-36 expression by KCs. IL-36 cytokines are secreted as precursors that require processing by neutrophil-derived proteases (elastase, cathepsin G, or protease 3) and KC-derived cathepsin S. Mature IL-36 cytokines bind to IL-36R on the KC cell surface, acting in an autocrine manner to further induce IL-36 expression. Moreover, they promote the production and release of pro-inflammatory cytokines (IL-1β, IL-12, IL-23, IL-6, and TNF-α), neutrophil chemokines (CXCL1, CXCL2, CXCL6, and CXCL8 [(IL-8)], lymphokines (such as CCL20, CCL3) and co-stimulatory molecules by DCs and Th1. They also induce T-cell proliferation, and cytokines secreted by infiltrating Th1 and Th17 lymphocytes further potentiate this inflammatory loop by inducing IL-36 expression and other pro-inflammatory mediator production by KCs. Mon: monocytes; pDC: plasmatic dendritic cell; TIR: Toll/interleukin-1 receptor.

## Data Availability

Not applicable.

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
