# Peer review of "Exploring the Role of IL-36 Cytokines as a New Target in Psoriatic Disease"

_ijms, 2021, doi:10.3390/ijms22094344_

Round 1

Reviewer 1 Report

The review is complete and well developed even if it could be improved by referring to the very recent manuscripts (Miura S, et al., Majumder S, et al., Wang WM et al., Caputo V et al.,). The title should be changed, as it is not suitable for a review. The role of IL36 as a possible therapeutic target needs to be emphasized more in the conclusion.  Furthermore, the authors must explain how IL36 can be a good therapeutic target for Psoriasis and PsA.

Author Response

We would like to express our thanks to the reviewers for their feedback on our manuscript, and we are grateful to have been given the opportunity to respond and resubmit a revised version based on this feedback.

Please find, itemized below, our responses to each of the reviewers’ comments. All revisions to the manuscript have been made using tracked changes and have been approved by all co-authors. 

Reviewer 1

The review is complete and well developed even if it could be improved by referring to the very recent manuscripts

We have now revised and included Wang WM et al. and Caputo V et al. works in our review.

The title should be changed, as it is not suitable for a review. 

Title has been changed to Exploring the role of IL-36 cytokines as a new target in psoriatic disease

The role of IL36 as a possible therapeutic target needs to be emphasized more in the conclusion.  Furthermore, the authors must explain how IL36 can be a good therapeutic target for Psoriasis and PsA.

The role of IL-36 has been further emphasized in the conclusion, and detailed explanation of its role in PsA and psoriasis has been added.

Reviewer 2 Report

The authors have performed a comprehensive review of the role of IL-36, its receptors, and receptor antagonist in psoriasis. I believe that it is a well-researched article that contributes to the comprehension of complicated molecular pathways. 

The authors have presented a very thorough review of the role played by IL-36 and IL-36 receptor in the pathogenesis of psoriasis, and in particular, pustular psoriasis. The current study is very comprehensive and well written and it summarizes a very recently discovered complicated and sometimes overlooked topic in the pathogenesis of psoriasis. The authors are very successful in teasing out the complex interactions of the different forms of IL-36 and its receptor. This could be enhanced by a few more details on the molecular roles of the receptor antagonist as well as IL-38. Overall, this manuscript is clear and very informative on a subject of high interest. I have the following suggestions: 1) Add a short paragraph dedicated to palmoplantar pustulosis (PPP). The latter is of particular interest since its association to psoriasis has been a matter of discussion, as well as representing a disease that has been resistant to the mainstream treatments for psoriasis. Generalized pustular psoriasis is rare whereas PPP is found much more often, and frequently in the absence of other lesions of psoriasis. Additionally, the majority of PPP has not been reported to have significant alterations of the IL36RN gene. See: • Takahashi T, Fujimoto N, Kabuto M, Nakanishi T, Tanaka T. Mutation analysis of IL36RN gene in Japanese patients with palmoplantar pustulosis. J Dermatol. 2017;44(1):80–3. • Wang TS, Chiu HY, Hong JB, Chan CC, Lin SJ, Tsai TF. Correlation of IL36RN mutation with different clinical features of pustular psoriasis in Chinese patients. Arch Dermatol Res. 2016;308(1):55–63. • Twelves S, Mostafa A, Dand N, Burri E, Farkas K, Wilson R, et al. Clinical and genetic differences between pustular psoriasis subtypes. J Allergy Clin Immunol. 2019;143(3):1021–6. • Mossner R, Wilsmann-Theis D, Oji V, Gkogkolou P, Lohr S, Schulz P, et al. The genetic basis for most patients with pustular skin disease remains elusive. Br J Dermatol. 2018;178(3):740–8. 2) Table 1 does not add any significant knowledge to the manuscript, since the main points are written in the main text. It can be removed. 3) Given the relevance of IL-36RA and IL-38 in the described pathway of IL-36, it would be useful information to include the cells that produce these two molecules.

Author Response

We would like to express our thanks to the reviewers for their feedback on our manuscript, and we are grateful to have been given the opportunity to respond and resubmit a revised version based on this feedback.

Please find, itemized below, our responses to each of the reviewers’ comments. All revisions to the manuscript have been made using tracked changes and have been approved by all co-authors.

Add a short paragraph dedicated to palmoplantar pustulosis (PPP). The latter is of particular interest since its association to psoriasis has been a matter of discussion, as well as representing a disease that has been resistant to the mainstream treatments for psoriasis. Generalized pustular psoriasis is rare whereas PPP is found much more often, and frequently in the absence of other lesions of psoriasis. Additionally, the majority of PPP has not been reported to have significant alterations of the IL36RN gene.

  • Takahashi T, Fujimoto N, Kabuto M, Nakanishi T, Tanaka T. Mutation analysis of IL36RN gene in Japanese patients with palmoplantar pustulosis. J Dermatol. 2017;44(1):80–3.
  • Wang TS, Chiu HY, Hong JB, Chan CC, Lin SJ, Tsai TF. Correlation of IL36RN mutation with different clinical features of pustular psoriasis in Chinese patients. Arch Dermatol Res. 2016;308(1):55–63.
  • Twelves S, Mostafa A, Dand N, Burri E, Farkas K, Wilson R, et al. Clinical and genetic differences between pustular psoriasis subtypes. J Allergy Clin Immunol. 2019;143(3):1021–6. • Mossner R, Wilsmann-Theis D, Oji V, Gkogkolou P, Lohr S, Schulz P, et al. The genetic basis for most patients with pustular skin disease remains elusive. Br J Dermatol. 2018;178(3):740–8.

We thank you for your suggestion, as we believe it adds interest to our work. We have now included information on PPP and the phenotype/genotype correlation. We have incorporated Takahashi et al, Wang et al, Twelves et al and Mössner et al studies.

Table 1 does not add any significant knowledge to the manuscript, since the main points are written in the main text. It can be removed.

Table 1 has been removed

Given the relevance of IL-36RA and IL-38 in the described pathway of IL-36, it would be useful information to include the cells that produce these two molecules.

Keratinocytes are the main producers of IL-38 and IL-36RA. This information has been included.

Reviewer 3 Report

I think that the authors summarized the role of IL-36 in psoriatic diseases well; however, I have some minor concerns.

(1) Are there any molecules which inhibits the function of IL-36 without IL-36a or IL-38 ?

(2) Could IL-36-inhibitors significantly reduce disease activity of psoriatic patients even after IL-17- or IL-23-inhibitors failed ?  With recruited patients in many ongoing clinical trials of IL-36-inhibitors, are IL-17-inhibitors failure or IL-23-inhibitors failure included ? 

(3) Is IL-36 expressed at enthesis with psoriatic arthritis patients ?

Author Response

We would like to express our thanks to the reviewers for their feedback on our manuscript, and we are grateful to have been given the opportunity to respond and resubmit a revised version based on this feedback.

Please find, itemized below, our responses to each of the reviewers’ comments. All revisions to the manuscript have been made using tracked changes and have been approved by all co-authors. 

Reviewer 3

I think that the authors summarized the role of IL-36 in psoriatic diseases well; however, I have some minor concerns.

Are there any molecules which inhibits the function of IL-36 without IL-36a or IL-38?

Molecules directed against IL-36y have been shown to inhibit inflammation (in vivo and in vitro) by blocking its interaction with IL-36R (Todorović V, Su Z, Putman CB, et al. Small Molecule IL-36γ antagonist as a novel therapeutic approach for plaque Psoriasis. Sci Rep. 2019;9:9089: )

Blocking IL-1RAcP has been studied in pre-clinical disease models. However, the common expression of IL-1RAcP in various cell and tissue types raises safety concerns regarding the systemic use of these antibodies. (Ågerstam H, Hansen N, von Palffy S, et al. IL1RAP antibodies block IL-1-induced expansion of candidate CML stem cells and mediate cell killing in xenograft models. Blood. 2016;128:2683–2693.)

Could IL-36-inhibitors significantly reduce disease activity of psoriatic patients even after IL-17- or IL-23-inhibitors failed?  With recruited patients in many ongoing clinical trials of IL-36-inhibitors, are IL-17-inhibitors failure or IL-23-inhibitors failure included? 

The rationale behind IL-36 inhibitors success after IL-17/IL-23 failure is the targeting of a different pathway.  Phase I spesolimab trial included biologic naïve patients.

Phase II and III trials of spesolimab and imsidolimab were recruiting bio-experienced patients but results have not been posted yet.

Is IL-36 expressed at enthesis with psoriatic arthritis patients?

We have not found published data regarding IL-36 expression at the entheses of PsA patients, but a recently published study reports IL-36a expression in PsA synovium, as mentioned in our document (Boutet MA, Nerviani A, Lliso-Ribera G, Lucchesi D, Prediletto E, Ghirardi GM, Goldmann K, Lewis M, Pitzalis C. Interleukin-36 family dysregulation drives joint inflammation and therapy response in psoriatic arthritis. Rheumatology (Oxford). 2020 Apr 1;59(4):828-838.).

Round 2

Reviewer 1 Report

The manuscript was carefully revised, taking into account the comments received.
The choice of the new title coincides with the aim of the paper.
A careful revision of the manuscript is recommended in order to correct small language and typing errors.